# Virtual Design of 3D-Printed Bone Tissue Engineered Scaffold Shape Using Mechanobiological Modeling: Relationship of Scaffold Pore Architecture to Bone Tissue Formation

**DOI:** 10.3390/polym15193918

**Published:** 2023-09-28

**Authors:** Adel Alshammari, Fahad Alabdah, Weiguang Wang, Glen Cooper

**Affiliations:** 1School of Engineering, University of Manchester, Oxford Road, Manchester M13 9PL, UK; fahad.alabdah@postgrad.manchester.ac.uk (F.A.); weiguang.wang@manchester.ac.uk (W.W.); 2Engineering College, University of Hail, Hail 55476, Saudi Arabia

**Keywords:** virtual design, tissue engineering, pore architecture, scaffold shape, cell behavior, mechanobiological modeling

## Abstract

Large bone defects are clinically challenging, with up to 15% of these requiring surgical intervention due to non-union. Bone grafts (autographs or allografts) can be used but they have many limitations, meaning that polymer-based bone tissue engineered scaffolds (tissue engineering) are a more promising solution. Clinical translation of scaffolds is still limited but this could be improved by exploring the whole design space using virtual tools such as mechanobiological modeling. In tissue engineering, a significant research effort has been expended on materials and manufacturing but relatively little has been focused on shape. Most scaffolds use regular pore architecture throughout, leaving custom or irregular pore architecture designs unexplored. The aim of this paper is to introduce a virtual design environment for scaffold development and to illustrate its potential by exploring the relationship of pore architecture to bone tissue formation. A virtual design framework has been created utilizing a mechanical stress finite element (FE) model coupled with a cell behavior agent-based model to investigate the mechanobiological relationships of scaffold shape and bone tissue formation. A case study showed that modifying pore architecture from regular to irregular enabled between 17 and 33% more bone formation within the 4–16-week time periods analyzed. This work shows that shape, specifically pore architecture, is as important as other design parameters such as material and manufacturing for improving the function of bone tissue scaffold implants. It is recommended that future research be conducted to both optimize irregular pore architectures and to explore the potential extension of the concept of shape modification beyond mechanical stress to look at other factors present in the body.

## 1. Introduction

Globally, approximately forty million large bone defects occur every year, and 5–15% of all fractures result in non-union or impaired healing [1]. However, large bone fracture treatment remains clinically challenging due to the requirements of cell differentiation, migration, and proliferation to facilitate healing. The most common treatment used in such cases is autologous bone grafting, which has several drawbacks, including the limited sources of bone graft tissue available and the need for further surgery [2]. The use of biomaterials, particularly in 3D-printed bone tissue engineered scaffolds, is now more commonly considered as an alternative to the use of bone grafts with human tissue [3,4].

Despite the positive outcomes observed with 3D-printed bone tissue engineered scaffolds [5], there are barriers to implementing the new bone tissue scaffolds developed in research laboratories for use in clinical purposes. Some of these confluence factors have been reported by Hollister [6], such as (1) the need for a more comprehensive understanding of scaffolding materials and design specifications; (2) the demand for a deeper comprehension of material deterioration in therapeutically relevant applications, particularly in relation to regulatory requirements; (3) the need to incorporate computational design techniques and manufacturing processes more effectively; and (4) the necessity of increased participation of the scaffold’s end users, namely users, in the design process in order to encourage surgical acceptance and application of biodegradable scaffolds [6]. The findings of Hollister [6] are related to the design processes of material selection, computer-aided design, and manufacture and user-centered design. So, improving 3D-printed bone tissue engineered scaffold design will enable better translation of these technologies into clinical practice. Since bone healing involves both mechanical and biological processes, determining the best design performance of bone tissue scaffolds is challenging. For instance, increasing the pore size of the scaffold may enhance the biological aspect, while dramatically weakening the scaffold’s stiffness and strength [7].

The functional design of components is well researched in engineering, and the relationships between function, shape, material, and manufacturing have been explained by Ashby et al. [8]. Wagoner Johnson et al. [9] and Shimojo et al. [10] drew out key requirements for 3D-printed bone tissue engineered scaffolds, which from now on we will refer to as ‘scaffolds’. In their work, they highlight the importance of biocompatibility, biodegradability, pore morphology, porosity, chemical properties, mechanical function, and processability. Furthermore, modifying environmental factors, such as changing mechanical stress with fixation [11], use of biochemicals using growth factors [12], and more recently, electrical field changes through stimulation [13], can also improve implant function [6]. Finally, many of these scaffolds would not have been created without the advances seen in additive manufacturing techniques, for example 3D printing [14], phase separation [15], two-step bulk polymerization [16], and solvent casting/particulate leaching [17]. Figure 1 is adapted from engineering design principles [8], and shows how implant function is directly affected by material, shape, and environmental factors, and indirectly affected by manufacturing through shape and material.

Design involves many stages, but the most commonly employed in tissue engineering is experimental feasibility. Pre-clinical and clinical studies which investigate the effect of scaffold design are commonly expensive and require a lot of time to investigate the bone regeneration process; for example, the mainstream in vitro cell test methodology generally takes 7 to 28 days [18,19], at least one month to run, and may take a further month to run tests and analysis. However, it is suggested that in silico approaches could enable exploration of a much wider range of scaffold design parameters and reduce time and monetary costs, as well as overcome ethical challenges by allowing the rapid identification of an optimal design. The currently investigated experimental methods are mainly implemented based on a trial-and-error approach [20]; this contrasts with most other engineering and medical components, which usually begin with design and virtual prototyping rather than practical experiments. Scaffolds do not replace the native bone during the regeneration process, instead providing an appropriate environment in mechanical and biological terms for the bone to regenerate. Currently, tissue engineered bone implant technologies are designed through a physical prototype method and practical experiments in vitro and in vivo. It is suggested that predictive healing computer models, i.e., virtual prototypes, may thus be developed that account for the cellular activities such as migration, proliferation, and differentiation, and support such regeneration, pre-operative schemes for customized bone scaffold designs that might encourage the greatest formation of bone during the regeneration process. Conducting a search on the research papers in the area of tissue engineering using the search terms of ‘scaffold material’ and ‘scaffold shape’ gives 40,651 articles and 7381 articles, respectively. However, from Figure 1, it is clear that shape (or scaffold geometry and architecture) is equally able to affect bone tissue implant function as material, yet there appears to be nearly six times less research in this area. This suggests that focusing research effort on scaffold shape and the use of in silico approaches could make additional functional gains and facilitate better bone regeneration.

Experimental studies have looked at different bone tissue scaffold implant architectures to observe their impacts on the bone regeneration process; such investigations have considered pore size [21,22,23,24], pore gradient [25], pore shape [26], and scaffold material [27,28], all of which at least partially impact the bone generation process. However, there are still some contradictory conclusions, such as the fact that while one study reported the minimum pore size allowing good oxygen diffusion to be 100 µm [29], other studies found good bone formation with scaffolds with pore sizes of around 25 µm [30]. Tissue engineered bone implant technologies are largely focusing on material innovations and regular architectures with microstructures that have interconnected porosity of around 300–800 µm pore size [21,22,23,30,31,32,33,34,35,36,37,38].

Previous researchers have employed methods to create mechanobiological (governing) rules to predict bone formation during bone healing [39,40,41,42,43]. Other validated in silico studies have used computer models based on these rules to investigate the bone regeneration process within scaffolds [44,45]. Previous in silico approaches have investigated various design parameters related to the long-term bone regeneration process, including assessing scaffold pore sizes [31], optimizing time-dependent mechanobiology-based topology for the design of tissue scaffolds [46], and developing a unique dual-porous scaffold (coarse and fine pores) and two models with controlled cubic pore sizes [47]. However, they have only investigated regular pore architectures, potentially due to the complexity of irregular architectures and the lack of experimental data on implemented irregular pore structure. The FE method provides a way to assess the stress and strain of scaffold architectures subjected to in vivo loading conditions, which when linked to agent-based models, can show such stimuli acting on the regenerating tissue [48]. It is suggested that using a virtual prototyping design framework to prototype implants, including both material and shape, in addition to other environmental factors (mechanical loading, electrical currents, biochemical environment), could create a tool to optimize implants for better bone regeneration.

Previous studies have indicated that considerable research efforts have been focused on material but relatively little on scaffold shape, evidenced by most scaffolds using a regular pore architecture throughout. Hence, the existing knowledge deficit regarding the relationship between material and shape can potentially be addressed by employing computational methods to assess the architectural characteristics of bone tissue engineering scaffolds. This approach would facilitate an initial wide exploration of designs in a virtual environment and the use of computational optimization, allowing researchers to concentrate their efforts on more promising design solutions, thereby minimizing the need for costly and time-consuming in vitro and in vivo experiments. This study proposes a virtual design framework to enable the development of scaffolds. It then introduces a multi-scale computer model that utilizes mechanobiological relationships to assess the impact of scaffold form on the process of bone regeneration, specifically pore architecture in this case study. The data from an experimental investigation conducted in vivo have been utilized to compare and validate the predictions made by the model. The investigation of scaffold pore gradients, both regular and irregular, has been conducted. This paper aims to demonstrate how different scaffold shape concepts (specifically scaffold architecture) can be evaluated as a virtual prototype to create more optimal bone tissue scaffolds. This study provides evidence that the architecture of scaffolds, particularly those with irregular scaffold pore gradients, has a significant impact on the functionality of scaffolds, leading to improved bone regeneration processes. The multi-scale model, which has been validated using in vivo data [49], has demonstrated its capacity to evaluate various scaffold architectures. As a result, it has promise for the evaluation and support of pre-clinical investigations.

## 2. Materials and Methods

### 2.1. Virtual Design Framework

A virtual design framework is proposed that will take input parameters of bone defect shape, biomechanical loading, and cell conditions and use these to simulate bone scaffold implant performance over time. Figure 2 shows the steps within the virtual design framework. The software is available for download in the Appendix A and an explanation of how to use the code is given in Appendix B. Specifically, inputs will consist of proposed implant shape (defect geometry and scaffold architecture), biomechanical loads exerted on the bone and implant (e.g., compressive load, bending moments, or other boundary conditions), and cell conditions (cell type, number and location, factors for scaffold material, and human subject metabolism). The mechanical stress environment is then evaluated using a FE model of the bone, fixation, callus, and bone tissue scaffold implant. The output of strain and hydrostatic stress on the bone tissue scaffold implant is then used in an agent-based model, which will calculate cell differentiation, cell migration, cell multiplication, and tissue formation (granulation, cartilage, bone, or tissue absorption) for that particular point in time. These outputs, including the structural and geometry of the new tissue formed, will then be inputted back into the model to enable calculation of the bone scaffold performance for the next time period. These steps are repeated to enable a time-dependent model of bone healing to be generated to evaluate the bone tissue scaffold in the virtual environment.

#### 2.1.1. FE Model

Within this study, a linear elastic FE model was constructed to evaluate the principal strain and hydrostatic stress of the bone scaffold implant. The FE model was developed using Abaqus/CAE 2022 (Simulia, Johnston, RI, USA). The minimal principal strains and hydrostatic stresses are calculated and used as inputs for a mechanobiological regulation model based on Claes et al. [39], as shown in Equation (1), in the agent-based model discussed in Section 2.1.2. 

Bone scaffold implant shape is case specific, but the parts of the model are common to all cases, and consist of bone, fixation, callus, and bone scaffold implant. The callus was assumed to be a homogeneous, linear elastic tissue (granulation tissue–amorphous solid) both occupying the bone scaffold implant’s pores during the initial phase of healing and surrounding the bone defect. 

Material properties for the FE model were taken from the literature [49,50,51,52,53,54,55] and are shown in Table 1.

#### 2.1.2. Agent-Based Model (Python Model)

Python programming language was used to create an agent-based model. Outputs of strain and hydrostatic stress from the FE model are used as some of the inputs for this agent-based model. The code will then define and simulate a callus part with a 50 µm spaced, three-dimensional grid. Every point in the lattice structure will be occupied by a single type of cell (mesenchymal stem cells (MSCs), fibroblasts, chondrocytes, or osteoblasts). The callus was initially filled with granulation tissue and MSCs were produced from the bone marrow cavity and the periosteum, which has been found to be rich with MSCs [56,57]. A share of 30% of the bone marrow cavity and the periosteum simulated points were hypothesized initially to be seeded with MSCs [58]. The bone regeneration model was then run iteratively to simulate tissue formation over a time period. A latency period of 7 days was implemented, which has been found to be the optimal period, after which the activity of the cells was hugely reduced [48]. Baseline rates of proliferation, apoptosis, differentiation, and MSC migration speed were used in previous in silico approaches, as shown in Table 2. Due to the lack of the biological activities in large bone defects and because the focus of this study is more on shape rather than material, the after-latency period cell activities of proliferation and differentiation rates mimicked the rates of an empty defect bone healing of a previously validated in silico model; see Table 2.

Cellular activity, specifically the differentiation procedure, follows the mechanobiological rules (S) proposed by Claes et al. [39] and the bone resorption area [59]; in turn, this is based on the hydrostatic stress and minimal principal strain, as given in Equation (1):(1)$$S=S(\gamma ,p)=\frac{\gamma }{a}+\frac{p}{b}$$
where γ is the minimal principal strain and p is the hydrostatic stress, a = 0.0375, b = 0.003 mm/s [60].

MSCs migrate randomly in six possible agent spaces and this process is repeated every iteration in 7 jumps to achieve a MSC migration velocity of 30 µm/h [61]; see Table 2. Proliferation of cells occurs randomly in six possible positions, only if these positions are not occupied by another cell phonotype. The differentiation of MSCs into fibroblasts, chondrocytes, and osteoblasts is determined using the mechanobiological theory aforementioned in Equation (1) based on hydrostatic stress and minimal principal strain. Every agent in the lattice is occupied by only one cell phenotype. The apoptosis process occurs for all tissue types at different rates [62]; see Table 2. New tissue formation material properties are updated after every three iterations to the corresponding element in the callus part in FE model. 

The FE model is updated with the new material properties with the new tissue formation (see Table 2) of each element in accordance with the mechanobiological regulation theory described in Equation (1) with an interactive update model. This update occurs every three iterations after calculating the average of every model element over the previous three iterations. Every iteration represents one day of recovery; see Figure 2. This is to reduce the computational cost, and its effect was negligible in the simulation predictions. 

**Table 2 polymers-15-03918-t002:** Cellular activities including proliferation, apoptosis, differentiation, and migration rates per day [44,61,62].

	Proliferation Rate per Day		Apoptosis Rate per Day	Differentiation Rate per Day		Migration Speed (µm/h)
	Baseline	After Latency Period		Baseline	After Latency Period	
**Stem Cells**	0.3	0.12	0.05	0.3	0.06	30
**Fibroblast**	0.275	0.11	0.05	-	-	-
**Chondrocyte**	0.1	0.04	0.1	-	-	-
**Osteoblast**	0.15	0.06	0.16	-	-	-

### 2.2. Model Validation

#### 2.2.1. Experimental Data from the Literature

In order to validate our model, experimental data were utilized from the literature, specifically in vivo experiments on the femurs of rodents [49]. Within the article, a 5 mm bone defect was cut in rats, and a PEEK fixation plate was placed facially with four screws, two of which were placed adjacent to the bone defect on both sides, and the other two were inserted into the extremities of the PEEK fixation [49]. They used a HA-PELGA scaffold as the bone implant in their study with a 3 mm diameter and a 3 mm length. The model simulated the natural bone growth by using previously validated models’ cellular activities’ rates, including migration speed, differentiation, apoptosis, and proliferation rates, as shown in Table 2 of Section 2.1.2.

#### 2.2.2. Virtual Design Framework Set Up

The bone, fixation, scaffold, and boundary conditions from the literature [49] were replicated in the virtual design framework.

*Material properties* of all the models utilized in this study were characterized as linear elastic materials, with their respective Young’s modulus and Poisson’s ratio values provided in Table 1. 

*The geometry* of the FE model is shown in Figure 3. The scaffold geometry was constructed based on the CAD illustration used in the experimental study from the literature [49]; the filament cross section was 0.4 mm and pore size was 0.8 mm; see Figure 3. The callus geometry was estimated based on the histology of the graft-guided bone data, using a Boolean method similar to that described in Section 2.1.1; see Figure 3. Our in silico study incorporates seven components, namely bone fixation, two cortical bones, two bone marrow parts, a scaffold, and a callus part. The length of the defect measured 5 mm. The bone fixation procedure involved securing a rectangular cube measuring 9 mm in length and 2 mm in width. Four screws were utilized, each possessing identical cylindrical geometries with a diameter of 0.5 mm and a length of 3.5 mm. The two cortical bones were observed to be structurally intact and exhibited a tube-like geometry, encompassing the two bone marrows. The bone marrows possessed a diameter of 2 mm and a length of 3 mm, while the cortical bones themselves had a thickness of 0.5 mm. The callus section of an arc geometry was designed with the intention of replicating the histological illustrations in the experimental study. Ultimately, this section will be removed from the scaffold utilized in the study [49]. The dimensions of the callus arc were measured to be 9 mm in length and 5 mm in diameter, with an overlap observed with the geometries of the cortical bone. The outer callus arc geometry remains consistent across all validations (Section 2.2) and investigated cases studies (Section 2.3). However, the inner geometry subtraction is contingent upon the specific scaffold geometry employed in each respective case.

*Mesh design* was implemented through an investigation of mesh sensitivity to predict the accuracy of the simulation, with the choice of mesh size optimizing the threshold between the results and computing efficiency. The converged mesh designs were 0.8 mm and 0.08 mm for the callus and scaffold, respectively. All the parts of FE model were meshed using linear quadratic tetrahedral elements of type C3D10. The bone fixation mesh size was 0.2 mm. Cortical bone and bone marrow mesh sizes were 0.42 mm and 0.35 mm, respectively.

*Boundary conditions* of the model were established assuming that the callus region and the pores of the scaffold were initially filled with granulation tissue. MSCs were then placed on both sides of the bone defect, specifically on the periosteum and within the bone marrow cavities. Previous studies have demonstrated that these areas are abundant in MSCs [56,57,62]. The implementation of biomechanical loading was based on an experimental study conducted on the femur bone of rats during gait [63]. A 17.7 N compression load was applied on the cortical bone on the proximal side [62], which is equivalent to six times bodyweight, simulating peak loading during gait [63]; the weight of the rat from the experimental data from the literature was 300 g [49]. A 5.7 N shear load was applied at the distal end of the bone in the model (mid-shaft of the femoral bone), which is equivalent to 10.7 times bodyweight, the maximum shear load which creates the maximum bending moment produced during gait [63]. The proximal end of the model (cortical bone and bone marrow regions) was constrained in all degrees of freedom using tie constraints. In order to maintain consistent displacement of the connecting nodes, tie constraints were employed to secure four bone fixation screws and the corresponding four holes in intact bone, encompassing both cortical and bone marrow regions.

*Solving* simulations took approximately 6–8 h for 112 iterations on a standard workstation computer. This is equivalent to a bone healing period of 16 weeks, with each iteration representing one healing day. The variation in job execution time in the FE model can be attributed to the number of processors utilized in each run. The cell migration process in Python is characterized by its long duration, primarily due to the randomly occurring movement of agents and the substantial population of agents occupied by MSCs, both of which contribute to further deceleration of the cell migration process.

*The agent-based* model execution process followed the same procedures outlined in Section 2.1.2. The rates of various cellular activities, including migration speed per day, proliferation, apoptosis, and differentiation rates, were comprehensively elucidated in Table 2. The coupling of the finite element model is performed iteratively, as outlined in Section 2.1.1. 

*Validation analyses* are processed and visualized using Python libraries. The outcomes of each iteration are recorded in Excel (CSV) files and preserved for subsequent analysis. Each cell type was assigned a corresponding integer number (e.g., osteoblast cells were assigned number 2, as shown in the Appendix A) in a CSV file for each healing day. Subsequently, these cells were processed for qualitative visualization and quantitative calculations. The Python libraries NumPy, pandas, and plotly.graph_objs were utilized to implement the lattice component for each day of the healing process. Subsequently, the resulting data were visualized with the bone cells were colored in gray [64]. The quantification of bone, cartilage, and fibrous volumes was achieved by determining the integer count of tissue points occupied by each of these components within a lattice structure, as specified in a CSV file, for each day of the healing process. These counts were then multiplied by the respective volume of each tissue point ($20^{-3}{\;\mathrm{mm}}^{3}$) to obtain the overall volume of each component.

### 2.3. Case Study: Scaffold Architecture Gradient Effect

To investigate the effect of the longitudinal pore gradients on bone regeneration, three separate scaffold architectures were designed, one regular and two irregular pore gradients. The geometry of the model was the same as in Section 2.2, apart from the bone scaffold implants. The material properties, validated mesh design, and boundary conditions were taken from Section 2.1 and Section 2.2, respectively. Material properties of the scaffolds assumed that the scaffolds possessed the same material properties as polymer−ceramic composites, with a Young’s modulus of 1000 MPa and a Poisson’s ratio of 0.3 [42,45]. 

Three scaffold designs, depicted in Figure 4, were chosen to investigate the impact of varying longitudinal pore gradients. All scaffolds were made using filaments and pores with a rectangular shape. All designs had the same radial dimensions of 0.25 × 0.25 mm and 0.2 mm for the pores and filaments, respectively. The filaments and pores were varied longitudinally for all designs. Design one, REG, has regular geometry, where all the filaments and pores have the same longitudinal dimensions of 0.2 mm and 0.3 mm, respectively. Design 2, IREG1, has irregular filament and pore geometry with smaller sizes on the edge of the defect, which increased linearly to the largest size at the center of the defect. Longitudinal pore size and filament size both ranged from 0.1 to 0.4 mm. Design 3, IREG2, has irregular longitudinal filament and pore geometry with larger sizes on the edge of the defect, which reduce linearly to the smallest size at the center of the defect. From the literature, bone tissue scaffold pore sizes range from 100 to 800 um for both in vitro and in vivo studies [38,65,66], so pore sizes were chosen to vary longitudinally between 0.1 and 0.4 mm for IREG1 and IREG2, which is within the range of the values from the literature. Longitudinal sizes ranged from 0.4 to 0.1 mm for both the pores and filaments. Case study analysis was conducted similarly to the validation analysis presented in Section 2.2.2.

## 3. Results

### 3.1. Creation of Virtual Design Framework

A framework to design the scaffold in a virtual environment was successfully created using an FE model and agent-based model. The framework input data can be edited for a variety of bone defects and scaffold geometries. The model is available for download from the Appendix A.

### 3.2. Model Validation Using In Vivo Experiment Results from the Literature

The developed in silico multi-scale computer model was verified against an in vivo experimental study from the literature [49], which considered a 5 mm defect fracture in a rat model implanted with an HA-PELGA scaffold and a PEEK bone fixation as the reference group. The results in Figure 5 show how the model was used to predict bone formation to examine the biomechanics behind the influence of scaffold design parameters. Experimental data from the literature and the computer model of bone formation are in agreement, which validates the virtual design framework.

As observed, the predicted bone formation at the rat femur defect site in our model was similar to the experimental results from the literature in all time periods from 4 to 16 weeks [49]. Specifically, at 4 weeks mean bone formation was 5.1 and 3.7 ${\mathrm{mm}}^{3}$ (1.4 ${\mathrm{mm}}^{3}$ difference) for our model and the experiment, respectively, which falls within the standard error mean deviation for the 4-week experimental result of ±1.5 ${\mathrm{mm}}^{3}$. At 16 weeks it was 18.9 and 22.5 ${\mathrm{mm}}^{3}\;$(3.6 ${\mathrm{mm}}^{3}$ difference) for our model and the experiment, respectively, which falls within the standard error mean deviation of the 16-week experimental result period of ±3.8 ${\mathrm{mm}}^{3}$. 

Additionally, the predicted bone formation from our model had higher density on the bone fixation side, which was similar to that observed in the experimental study. Our model also showed bone formation both inside and on the surface of the scaffold, which was also observed in the experiment. These values of differences and general bone growth positions gave confidence that the model was in agreement with the experiment. The predicted bone formation of the 5mm rat femur defect from our simulation was within the range of measured bone formation of the in vivo experiment from the literature [49] for all the time periods of 4, 8, 12 and 16 weeks. Figure 5 shows the bone formation of both the in vivo experiment from the literature [49] and the results of the model developed in our virtual design framework. The predicted bone formation was shown to be denser in the bone fixation site, as shown in the experimental study. 

### 3.3. Case Study Results: The Influence of Longitudinal Pore Gradients

#### 3.3.1. Bone Formation

Figure 6 shows the graphical representation of bone volume changes from 4 to 16 weeks for three different scaffolds with different pore gradients, REG, IREG1, and IRG2. The results of bone formation show differences between the different scaffold designs. Specifically, they show that the irregular pore gradient scaffold, IREG2, produced more new bone tissue growth in all time periods. In comparison to the regular pore scaffold, REG, the irregular pore scaffold, IREG2, produced 15.66% more bone over the healing period. 

During the initial four-week period, the bone volumes of IREG2 and REG were predicted to have a maximum difference of 1 mm^3^. However, in the final iteration after 16 weeks, the observed difference exceeded 4 mm^3^, as depicted in Figure 6. IREG2 emerged as the most favorable design among the cases, as it consistently yielded the highest bone volume throughout the entire 16-week healing period. In the initial 4- and 8-week periods, REG exhibited a greater bone volume compared to IREG1. However, it is anticipated that during the subsequent 8-week timeframe (12 and 16 weeks), IREG1 will demonstrate a higher propensity for bone formation.

#### 3.3.2. Cartilage and Fibrous Formation

Figure 7 shows cartilage and fibrous tissue formation from 4 to 16 weeks for the three scaffolds. A conflict in fibrous growth between IREG1 and REG was observed. The fibrous formation at the end of 16 weeks was found to be largest in REG, measuring 32.1 mm^3^. In comparison, IREG1 and IREG2 had fibrous formations measuring 28.5 mm^3^ and 25.7 mm^3^, respectively. The volumes of cartilage exhibited variability across all cases. During the 4-week healing period, it was observed that IREG2 exhibited the greatest quantity of cartilage formation, measuring 0.4 mm^3^. In contrast, IREG1 and REG demonstrated lower levels of cartilage generation, measuring only 0.16 mm^3^ and 0.13 mm^3^, respectively. In contrast, after a healing period of 16 weeks, the IREG2 group exhibited the lowest quantity of cartilage at 1.8 mm^3^, whereas the REG group demonstrated the highest amount of cartilage at 2.1 mm^3^. The IREG2 group ranked second with a marginal difference at 2.1 mm^3^. Evidently, IREG2 and REG exhibited their maximum values at the 8-week mark of the healing period, subsequently declining until the conclusion of the healing period. In contrast, IREG1 demonstrated continuous growth throughout the entire healing period.

## 4. Discussion

In this research, a virtual design framework was created to enable the optimization of synthetic polymer bone tissue scaffold implants. The model was validated against an animal model and showed good agreement in bone volume formation at all time periods from 0 to 16 weeks. The framework was then used to demonstrate the importance of scaffold architecture relationship to implant function. Specifically, it showed that modifying longitudinal pore gradients from regular to irregular will affect bone formation over time. Gradients which are wider nearer the native bone ends and become smaller towards the center (IREG2) of the tissue scaffold implant showed 83% and 33% greater bone volume after 4 weeks and 10% and 17% greater bone growth after 16 weeks than IREG1 and REG, respectively. This demonstrates that implant shape, in particular pore architecture, is an important design factor to improve bone tissue scaffold performance, as proposed by Figure 1.

Our study presented a multi-scale computer model to simulate bone healing utilizing bone scaffold implants with daily biological and biomechanical variations throughout the healing period. Other researchers have also produced mechanobiological models but they have not implemented this as a design tool [39,40,50]. Specifically, to the author’s knowledge, this is the first study to investigate the influence of scaffold shape design parameters, including irregular scaffold architectures (longitudinal pore gradient effect), using a mechanobiological regulatory bone regeneration model. 

The findings of our computational study indicate that scaffold architecture primarily influences bone growth (implant function in Figure 1). The IREG2 design demonstrated the best characteristics from the limited designs within the case study, resulting in the highest bone volume compared to the other designs. There are time-related bone growth differences between IREG1 and REG, with the former giving larger bone growth in the earlier period and the latter giving more bone growth in the later period. This indicates that bone growth rates may be modulated through pore architecture design. Further work in this area would be needed to fully understand this functional potential. Bone formation originates from the bone marrow and periosteum, both of which contain a significant number of MSCs [56,57]. During the initial phase of bone formation, it was observed that larger pores of IREG2 (measuring 0.4 × 0.2 mm) resulted in the highest bone volume. The second largest pore size, denoted REG (measuring 0.3 × 0.2 mm), produced less bone volume compared to IREG2 but more than IREG1, which had the smallest pore sizes at the extremities. These observations occurred during the first 8 weeks of bone differentiation. This finding is consistent with prior research [67,68], as illustrated in Figure 6. During the second 8-week period (9–16 weeks), despite the fact that IREG2 exhibited a smaller pore size in the central region of the scaffold, it remained dominant in terms of higher bone volumes compared to REG and IREG1. This phenomenon may be attributed to the increased formation of bone during the initial 8-week period, which subsequently contributed to improved mechanical support and larger cell numbers throughout the remaining healing process. The significance of scaffold architecture in bone growth may be heightened by the presence of a critical defect (in this case 5 mm). By reconfiguring the dimensions of scaffold filaments and pores within the defect size, such as employing larger filaments at the extremities during bone growth initiation, improved mechanical support and enhanced vascularization due to larger pore sizes for nutrition could be achieved. Previous research has indicated that larger pore sizes are correlated with superior vascularization [67,68]. According to Luca et al., it has been reported that gradual pore gradient scaffold designs from large to small are more effective in promoting gradual osteogenic differentiation of MSCs [69]. Furthermore, during the subsequent eight-week period, the findings indicated that the bone volume in IREG1 surpassed that of REG. Specifically, there was an increase in pore size at the center of the scaffold in IREG2, allowing more cell activity, while REG exhibited no alteration in pore dimension (smaller pore sizes in the center than IREG2).

Previous research has demonstrated that scaffold architecture significantly impacts the osseointegration during in vivo studies [26,70]. Specifically, the impact of scaffold pore shape and interconnectivity on bone growth in large bone defects has been well documented [26,71]. Although these studies have demonstrated that pore size has a significant impact on bone growth, they have focused on regular pore structures and have not explored the design space of irregular pore architectures. This means that shape remains largely unexplored in research in comparison to material, growth factors, and fabrication methods. Given the large design space of bone tissue scaffold implant shape and the intricate relationships between shape and mechanobiological function, it may be more suitable to explore this research area using virtual modeling and optimization methods (such as artificial intelligence) to narrow the design space rather than using just experimental techniques. Nevertheless, our computational simulation method aligned with several prior in vivo investigations regarding the impact of gradient pores on bone growth [25,67,68]. It has also been validated against an in vivo experiment from the literature [49]. Additionally, our approach offered enhanced control over the design parameter investigated (pore architecture) to enable the collection of detailed quantitative data on various bone tissues. In addition to in vitro and in vivo investigations, in silico methodologies, particularly multi-scale models, have demonstrated their efficacy as virtual predictive tools for assessing various scaffold parameters. This has the potential to decrease the number of in vivo trials, thereby reducing both time and costs in accordance with the three Rs approach (Replacement, Reduction, and Refinement). However, this in silico model may exhibit limitations in accurately simulating certain processes, such as the impact of surface properties on the growth of bone cells. Essentially, it is only simulating at a micro level, and not at a nano level. It is plausible that changing pore sizes and filament sizes may slightly change surface properties on the filaments during practical manufacturing. However, this will be very minor in comparison to surface modification, which could be achieved through material change or secondary processes. 

Several studies have previously implemented the incorporation of local mechanics in the callus healing region to consider the formation of bone tissues during the soft and hard callus phases [50,62,72]. Impressive advancements have been made in the field of bone healing prediction. However, it is worth noting that the ultimate stage of remodeling was not fully predicted, leading to the erroneous assumption that the bone would persist entirely within the bone marrow cavity by the end of the simulation. Isaksson et al. and Byrne et al. have effectively conducted simulations of the concluding phase of bone remodeling [42]. However, their investigations have been limited to empty bone defects and have not been subjected to experimental validation. Isaksson et al. employed a rat model, while Byrne et al. utilized a human tibia defect. Prier-Metz et al. have conducted excellent research employing agent-based modeling in conjunction with finite element modeling to study various species (specifically, sheep and rat models) in both empty conditions and with the aid of scaffolds, with the aim of simulating the entire bone healing process [31,44,45,73]. These studies have examined the impact of different scaffolds’ material properties and different pore sizes, but with regular scaffold architectures, on the bone regeneration process. However, they have not explored scaffold shape, specifically the full design space of irregular bone tissue scaffold architectures. 

Our study has several limitations, which include: The predefined callus shape was created in silico and filled with granulation tissue, whereas no callus formation was observed in vivo. This is similar to many other mechanobiological models mentioned above [31,44,45,73], but one recent study modeled the callus behavior [74]. Nevertheless, the callus part was subjected to a biomechanically stressed environment, as determined by the mechanobiological model proposed by Claes et al. [39]. This mechanobiological formula makes it possible to predict the formation of various tissue types in callus areas, which may not be detectable in vivo with current measurement methods.No revascularization process of the defect was included in this simulation, despite the fact that other studies have found this to be a concerning issue in large bone defects [75,76,77,78]. In this simulation, this was mitigated by forming bone only in suitable biological and mechanical environments. This may have a slight impact on the results of the case studies focused on the effect of the longitudinal pore gradients.No scaffold degradation was included in the model. However, in this study Young’s modulus and the Poisson ratio for the three scaffold cases were 1000 MPa and 0.3, respectively, which are within the range of polymer−ceramics composites used in bone regeneration applications [61,79], which commonly use polycaprolactone (PCL), poly(l-lactide) (LPLA) or poly(lactic acid) (PLA) as the base polymers of the bone tissue scaffold implant. Lam et al. [80] reported that PCL scaffolds have no significant degradation and little effect on bone regeneration for periods of six months or less. Similar low degradation is reported for LPLA and PLA [81,82].Fixed relationships for cell behavior was a limitation of this model. Cell fates, migration, and multiplication are based on fixed relationships with the stress environment without any statistical variation. In an in vitro or in vivo experiment, statistical changes would be present. However, if these statistical changes were included the simulation would become more computationally expensive.

Further work is required to fully validate mechanobiological models, which would be aided by more open-source data on results from experimental studies (in vitro and in vivo) that give full information on bone tissue formation, boundary conditions, and bone defect details. 

## 5. Conclusions

In conclusion, this study produced a novel virtual design framework which used a multi-scale mechanobiological model by integrating Python coding with lattice tools and Abaqus/CAE 2022 (Simulia, Johnston, RI, USA) to examine the impact of irregular scaffold architecture, such as longitudinal gradient pores, on the bone regeneration process in large bone defects. This model was validated by in vivo experiments from the literature, and is available to download from Appendix A. It can be used as a virtual design framework to design and optimize bone tissue scaffold implants. It will enable reduction in the design space to focus more expensive and time-consuming in vitro or in vivo experimental efforts on more promising scaffold solutions.

This study shows that shape is an important factor in bone scaffold implant design. This is similar to the relationship proposed in engineering design, as shown in Figure 1; in this case, scaffold architecture directly affects implant function (regenerated bone), with irregular pore gradients yielding up to 17–33% more bone growth than regular pore architectures studied in the optimized case studies. Further work is required to extend mechanobiological models beyond mechanical stress/strain stimulus to include more of the complex aspects found during in vivo bone healing conditions.

## Figures and Tables

**Figure 1 polymers-15-03918-f001:**
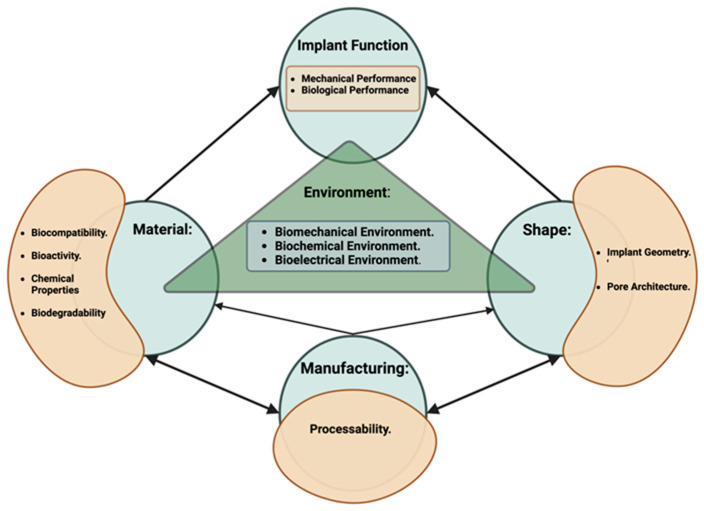
Relationship of material, shape, manufacturing, and environment on implant function; concepts were taken from Ashby et al. [8].

**Figure 2 polymers-15-03918-f002:**
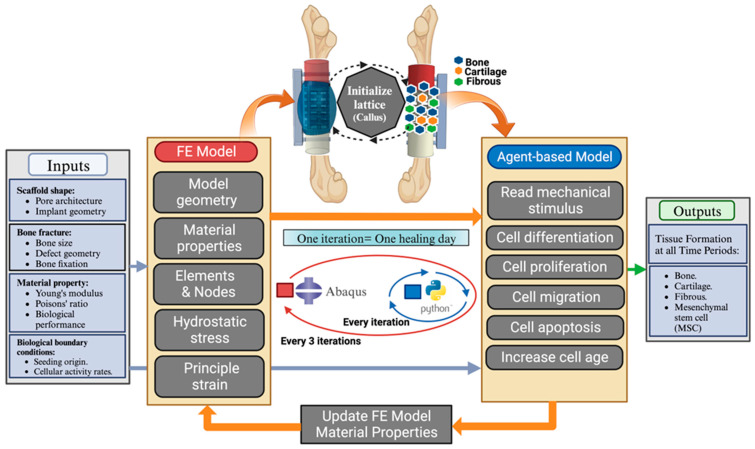
Virtual design framework containing a multi-scale mechanobiological model combining finite element analysis with a cell agent-based model.

**Figure 3 polymers-15-03918-f003:**
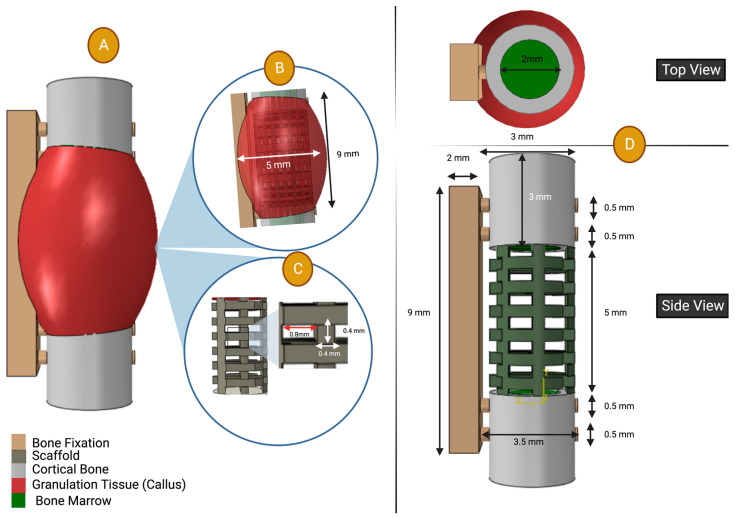
Description of the finite element model geometry: (**A**) Full geometry of the FE of the 5 mm rat femoral segmental defect regeneration [49]. (**B**) Callus geometry. (**C**) Scaffold architecture (square section filaments of 0.4 × 0.4 mm and line spacing of 0.8 mm). (**D**) Dimensions of the finite element model used in validation (Section 2.2) and the case study (Section 2.3) with details of cortical bone, bone fixation with four screws, and bone defect size. Note that the scaffold architecture does vary in each section.

**Figure 4 polymers-15-03918-f004:**
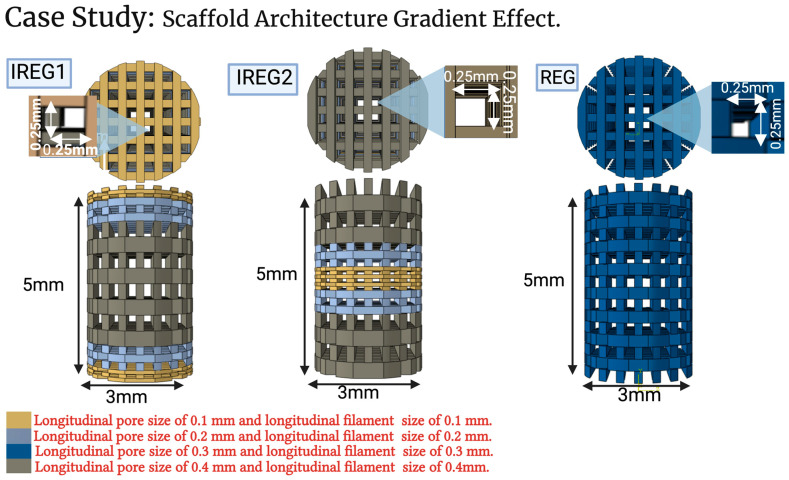
Scaffold designs with different pore architectures used for the case study showing longitudinal and radial views. IREG1 presents the scaffold with gradients which are smaller nearer the native bone ends and become wider towards the center. IREG2 presents the scaffold with gradients which are wider nearer the native bone ends and become smaller towards the center. REG presents the scaffold with all filament and pore sizes the same.

**Figure 5 polymers-15-03918-f005:**
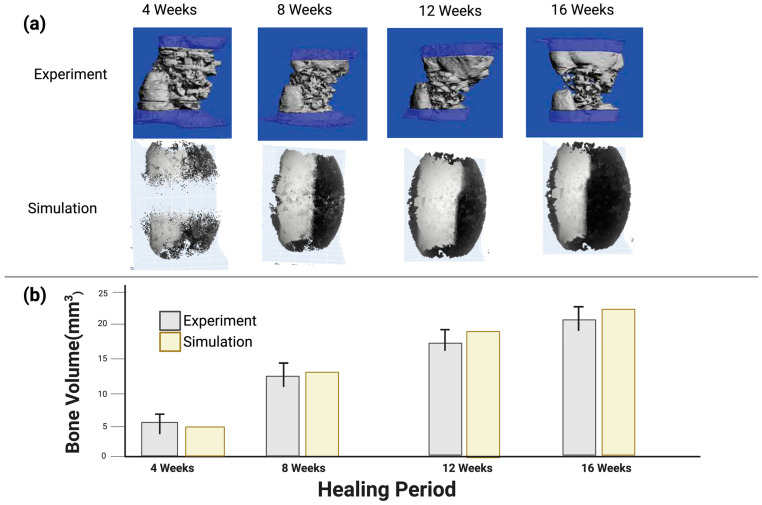
(**a**) Graphical outputs of the experiment and the simulation: the top row shows the bone formation histology from the experimental study from the literature [49], and the middle row shows the simulation bone prediction images. (**b**) Graph showing quantitative outputs from the experiment and the simulation of the regenerated bone volume for time periods from 4 to 16 weeks.

**Figure 6 polymers-15-03918-f006:**
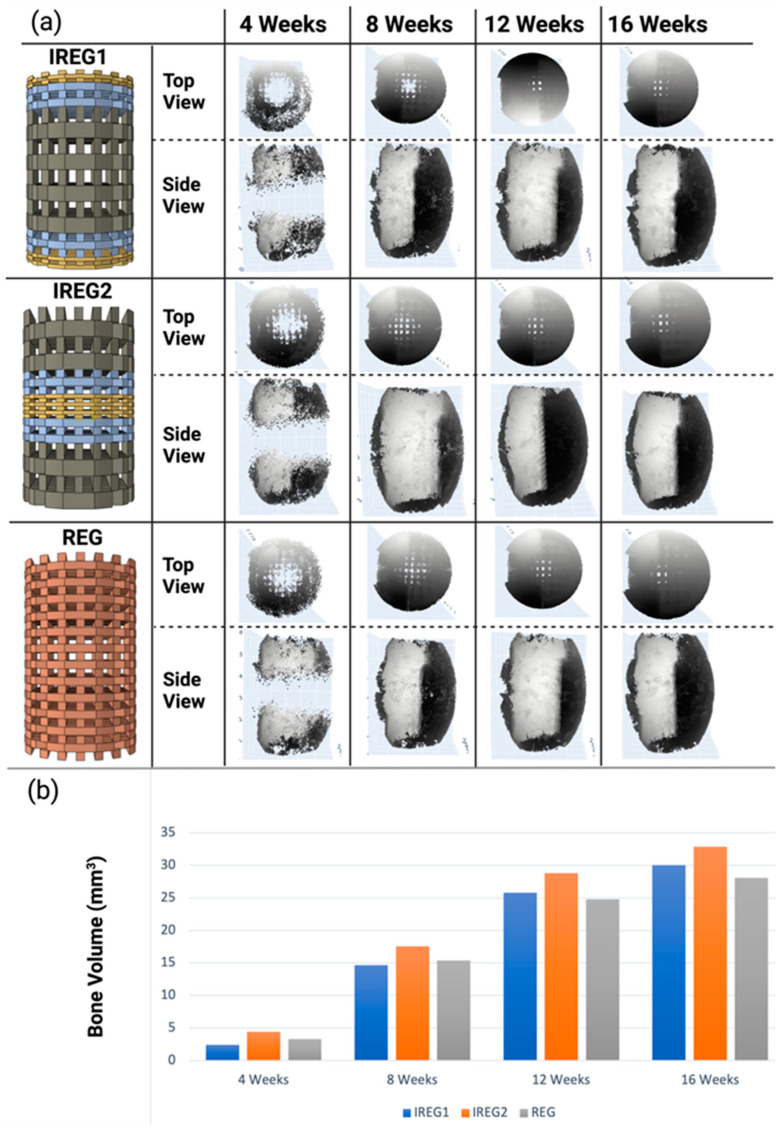
(**a**,**b**) Graphical and quantitative outputs of bone volume formation from the case study of scaffolds with different pore architecture designs: (**a**) graphical image of the bone formation predicted for the three scaffold architectures; (**b**) bar chart showing the bone volume for the three scaffold architectures at time periods from 4 to 16 weeks.

**Figure 7 polymers-15-03918-f007:**
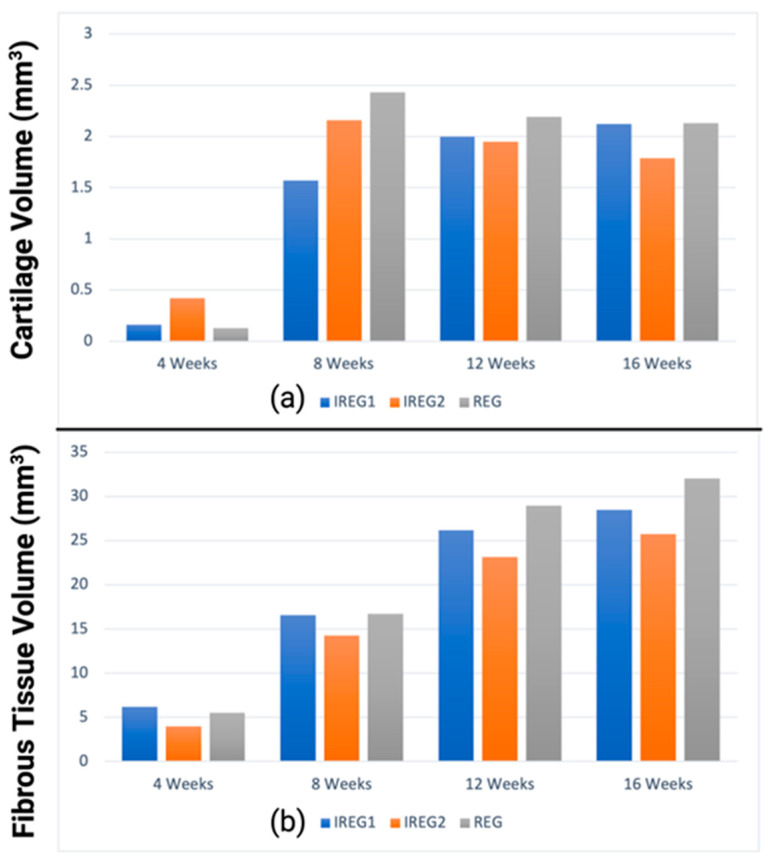
Graphs showing tissue formation for the three scaffold architectures in the case study for time periods from 4 to 16 weeks: (**a**) cartilage formation predicted from the simulation; (**b**) fibrous tissue formation predicted from the simulation.

**Table 1 polymers-15-03918-t001:** Material properties used in the mechanobiological computer model [49,50,51,52,53,54,55].

Material	Young’s Modulus (MPa)	Poisson’s Ratio
Granulation tissue	0.2	0.167
Fibrous tissue	2	0.167
Cartilage	10	0.3
Cortical bone	8000	0.3
Bone marrow	2	0.167
HA-PELGA scaffold	350	0.3
Polyether-ether-ketone (PEEK) fixation	3800	0.36

## Data Availability

All data for this research is either shown in the paper, available from reference [49] or could be replicated from the code in the Appendix A.

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
