# Peer review of "Virtual Design of 3D-Printed Bone Tissue Engineered Scaffold Shape Using Mechanobiological Modeling: Relationship of Scaffold Pore Architecture to Bone Tissue Formation"

_polymers, 2023, doi:10.3390/polym15193918_

Round 1

Reviewer 1 Report

The manuscript reports on the investigation of the relationship between pore architecture and bone formation through the use of a model. It is a very interesting manuscript for a preliminary study in this area.

 The following have to be addressed before the manuscript can be considered.

 1. The authors found that irregular pore gradient are better than regular pore gradient architecture for bone formation. What is the difference in size between the regular and irregular pore architecture?

 2. Can the irregular gradient impact on the surface properties of the scaffolds and thus on bone cell growth? Can the authors discuss this parameter.in section 4.

 3. The results in Figure 7 have no statistical analysis.

 4. Can the authors explain the size of the pores chosen in the models? Did they use the size/area of bone cells to create the model?

How does the area of bone cells compare to the chosen area of the pores?

 5. Did the authors use only 1 in vivo study to validate their model? Did the model mimic the natural bone being treated in the in vivo experiment?

Reviewer 2 Report

Please consider describing this model's limitations using a table or chart.

Justify how this model fits to mimic  native tissue  

Round 2

Reviewer 1 Report

The authors have addressed all queries.